# Detection and Segmentation of Radiolucent Lesions in the Lower Jaw on Panoramic Radiographs Using Deep Neural Networks

**DOI:** 10.3390/medicina59122138

**Published:** 2023-12-09

**Authors:** Mario Rašić, Mario Tropčić, Pjetra Karlović, Dragana Gabrić, Marko Subašić, Predrag Knežević

**Affiliations:** 1Clinic for Tumors, Clinical Hospital Center “Sisters of Mercy”, Ilica 197, 10000 Zagreb, Croatia; mario.rasic@kbcsm.hr; 2Faculty of Electrical Engineering and Computing, University of Zagreb, Unska Ulica 3, 10000 Zagreb, Croatia; mario.tropcic@fer.hr; 3Department of Maxillofacial and Oral Surgery, Dubrava University Hospital, Avenija Gojka Šuška 6, 10000 Zagreb, Croatia; soricpjetra@gmail.com; 4Department of Oral Surgery, School of Dental Medicine, University of Zagreb, Gundulićeva 5, 10000 Zagreb, Croatia; dgabric@sfzg.hr

**Keywords:** deep learning, panoramic radiography, radiolucent lesion, artificial intelligence

## Abstract

*Background and Objectives*: The purpose of this study was to develop and evaluate a deep learning model capable of autonomously detecting and segmenting radiolucent lesions in the lower jaw by utilizing You Only Look Once (YOLO) v8. *Materials and Methods*: This study involved the analysis of 226 lesions present in panoramic radiographs captured between 2013 and 2023 at the Clinical Hospital Dubrava and the School of Dental Medicine, University of Zagreb. Panoramic radiographs included radiolucent lesions such as radicular cysts, ameloblastomas, odontogenic keratocysts (OKC), dentigerous cysts and residual cysts. To enhance the database, we applied techniques such as translation, scaling, rotation, horizontal flipping and mosaic effects. We have employed the deep neural network to tackle our detection and segmentation objectives. Also, to improve our model’s generalization capabilities, we conducted five-fold cross-validation. The assessment of the model’s performance was carried out through metrics like Intersection over Union (IoU), precision, recall and mean average precision (mAP)@50 and mAP@50-95. *Results*: In the detection task, the precision, recall, mAP@50 and mAP@50-95 scores without augmentation were recorded at 91.8%, 57.1%, 75.8% and 47.3%, while, with augmentation, were 95.2%, 94.4%, 97.5% and 68.7%, respectively. Similarly, in the segmentation task, the precision, recall, mAP@50 and mAP@50-95 values achieved without augmentation were 76%, 75.5%, 75.1% and 48.3%, respectively. Augmentation techniques led to an improvement of these scores to 100%, 94.5%, 96.6% and 72.2%. *Conclusions*: Our study confirmed that the model developed using the advanced YOLOv8 has the remarkable capability to automatically detect and segment radiolucent lesions in the mandible. With its continual evolution and integration into various medical fields, the deep learning model holds the potential to revolutionize patient care.

## 1. Introduction

Panoramic radiography is one of the most indicated radiographic examinations because it provides two-dimensional information about the teeth and the maxillofacial skeleton [1]. Over 400 million radiographs are performed in the United States every year [2]. According to the U.S. Food and Drug Administration, 150 million of those are panoramic radiographs, and with the increased use of CT and MR scans, the average radiologist must now interpret one image every three to four seconds in an eight-hour workday [3]. Computer-aided diagnosis (CAD) has been utilized to help radiologists with that workload, but conventional CAD requires the extraction of relevant features or patterns within the images using various algorithms and techniques, which is again a time-consuming task. The development of deep convolutional neural networks (CNN) has revolutionized various fields, particularly computer vision, due to their ability to automatically learn hierarchical representations of data. CNN can accomplish various tasks, including image classification, object detection, and segmentation [4,5,6,7,8]. A deep learning CNN consists of three layers: a convolutional layer, a pooling layer and a fully connected (FC) layer. The convolutional layer is the fundamental building block of CNNs and is responsible for learning features from input data. It operates by applying a set of filters to the input data through a process called convolution. Pooling layers are interspersed between convolutional layers and serve to downsample feature maps while retaining essential information. They help manage computational complexity, reduce overfitting, and make the learned features more invariant to small translations or distortions in the input data. At the end of a CNN architecture, the fully connected layer utilizes the learned representations from previous layers to produce the final output of the neural network. While found at the end of many neural network architectures, their presence and size can vary, depending on the specific design of the network and the task it is designed to solve [9]. Their effectiveness in learning hierarchical representations from raw data has made CNNs a cornerstone of modern artificial intelligence and computer vision systems. Consequently, these breakthroughs have effectively surpassed the conventional limitations of CAD.

Using CNN as a part of deep learning has been rapidly progressing, but there are only a few studies on the automatic detection of odontogenic cysts and tumors in the jaw on panoramic radiographs [10,11,12,13]. Most of the application has been limited to the classification of teeth and diagnosis of dental caries [14,15].

The cysts and tumors usually do not cause symptoms until they start deforming surrounding anatomical structures with their growth, or until an inflammation occurs [16]. Because of that, they are mostly diagnosed by accidental radiological findings which leads to delayed diagnosis and subsequent poor treatment outcomes. Early diagnosis is the key for oral and maxillofacial surgeons to plan appropriate treatment and ensure the best possible outcomes for patients [17].

Therefore, the purpose of this study was to automatically detect radiolucent lesions in the lower jaw by developing a deep learning model based on new, state-of-the-art data augmentation and real-time object detection, You Only Look Once (YOLOv8).

## 2. Materials and Methods

This study was approved by the ethics committee of the Clinical Hospital Dubrava (2023/2103-01) and the ethics committee of the University of Zagreb School of Dental Medicine (05-PA-30-16-3/2023) and was performed in accordance with the tenets of the Declaration of Helsinki.

### 2.1. Patients Selection

Patients were selected from the imaging database of the Clinical Hospital Dubrava and the School of Dental Medicine. A total of 200 panoramic radiographs taken from 2013 to 2023 of patients who visited those two institutions were obtained. Panoramic radiographs were searched in the hospital information system (IN2 group, Zagreb, Croatia) according to the classification and terminology from the International Statistical Classification of Diseases and Related Health Problems.

The inclusion criteria were the presence of a radiolucent lesion in the lower jaw and histopathologic verification of the diagnosis. These radiographs included 226 radiolucent lesions (Table 1). Radicular cyst, ameloblastoma, odontogenic keratocyst (OKC), dentigerous cyst and residual cyst were the included diagnoses (Figure 1).

As a control group, we prepared 100 normal panoramic radiographs. Panoramic radiographs were obtained from adult patients and only one radiograph was used per patient.

The digital panoramic radiographs were obtained using CRANEX 3D (Planmeca OY, Helsinki, Finland) and CRANEX D (Soredex, Tuusula, Finland).

### 2.2. Preparation of the Imaging Data Sets

The digital panoramic radiographs were carefully retrieved from the image database and saved in the Joint Photographic Experts Group (.JPEG) format, maintaining a resolution of 2776 × 1480 pixels—the highest attainable quality produced by the institutions’ panoramic X-ray machine. Our primary emphasis has revolved around the process of detection and segmentation of lesions, a subject matter that has remained largely unexplored within previously published papers. This compelled us to seek out the utmost level of resolution available to effectively tackle this segmentation challenge with precision and clarity. Each radiograph was labeled manually by drawing the cortical margin and coloring the internal part of the radiolucent lesion with red color. It was performed by a radiologist and an oral and maxillofacial surgeon using the GNU image manipulation program (GIMP) which offers a comprehensive set of tools for image editing, including selection tools, paintbrushes, pencils, airbrushes, cloning, and healing tools. It supports a wide range of file formats for both importing and exporting images, including popular formats like JPEG, PNG, GIF, TIFF and PSD, ensuring compatibility with different sources and outputs. GIMP is a powerful tool that also supports layers and masks, allowing radiologists to stack different elements of an image and enabling non-destructive editing by selectively applying changes to specific areas. The radiologist used brightness, contrast, and color balance adjustments for preparing the medical images. Furthermore, during the manual labeling of lesions, different layers were utilized so that not even one pixel would be missed in the annotation process.

### 2.3. Annotation of Images

For each image within the dataset, YOLO labels were generated based on the specific task at hand. Detection labels encoded precise coordinates of the bounding box (center, relative height, relative width) encompassing the lesion, enabling accurate object localization. Meanwhile, segmentation labels contained the relative pixel positions along the lesion margin.

### 2.4. Data Augmentation

Due to the scarcity of panoramic images, we performed data augmentation to increase the diversity of our deep learning training dataset. This involved applying techniques such as translation, scaling, rotation, horizontal flipping and mosaic effects to each batch during the training process. Translation involves moving the image in a certain direction by a random value horizontally or vertically. By applying translation during training, we enable the model to learn to recognize objects regardless of their position in the image. Rotation is the process by which the image rotates around its center or some other point by a certain angle. During training, we applied rotation so that the model could learn to recognize objects from different angles. Applying scaling involves changing the size of an image, whether it is reducing or enlarging. The idea of scaling is to create the illusion that the object is closer or farther away from the observer. By using scaling during training, the model is enabled to better recognize objects of different sizes. Mosaic is an augmentation technique that focuses on creating new images for training by combining several existing images. The newly created image has the appearance of a mosaic composed of smaller images. The purpose of creating a mosaic is to increase the diversity and complexity of the training data set. By merging multiple images into one, the model learns to deal with situations in which multiple objects appear in different contexts and complex backgrounds. In this way, the generalization capabilities of the model are improved. In addition to the methods already mentioned, YOLO also uses additional image augmentation techniques which are not controlled via hyperparameters but are directly applied to all images. These techniques are implemented using the “albumentations” software library of the Python language. One of these methods is “Blur”, which adds noise to images using a convolution operation with a kernel of a given size. This technique applied to images simulates different turbidity levels to improve model robustness. “MedianBlur” is also used to add noise within images where it is used as the median of surrounding pixels. Another method used is “ToGray”, which converts images to their black-and-white counterparts; otherwise, this technique has no effect on the radiological images with respect to their characteristics. The last method used is Contrast Limited Adaptive Histogram Equalization, an image processing technique that improves local image contrast to improve object visibility.

Notably, it is worth emphasizing that the utilization of these augmentation methods was primarily used to increase the diversity rather than to expand the dataset size with new examples.

### 2.5. Model

We have employed the YOLOv8 deep neural network to tackle our detection and segmentation objectives. YOLOv8 is an open-source model built and maintained by the Ultralytics team and distributed under the GNU General Public License. It has the ability to predict bounding boxes and segmentation masks at the same time. YOLOv8 provides a range of five model sizes: nano, small, medium, large and extra-large. In our research, we employed the large model. It consists of 401 layers and a total of 45,939,903 parameters which enables it to effectively deal with complex segmentation tasks. A simplified illustration of the model’s structure is depicted in Figure 2. The model is composed of two key components: the “backbone” and the “head”. The backbone is responsible for extracting crucial features from input images which are then used by subsequent layers to perform specific tasks. The backbone is essentially a series of convolutional layers (Conv) that conducts two-dimensional convolution operations, implements batch normalization and uses SiLU as an activation function. At the end of the backbone, the spatial pyramid pooling layer (SPPF) aggregates features of different scales into a fixed-size feature map.

The final part of the neural network, the “head”, is responsible for making specific predictions based on the extracted features. It uses concatenating and upsampling layers to increase the resolution of the feature maps and Coarse-to-Fine (C2f) layers to combine high-level features with contextual information to improve predictions. Detection modules use a series of convolutional layers to predict bounding boxes and class probabilities on feature maps of different scales. By combining these predictions, we obtain the final result.

The dimensions within each layer are contingent upon the input image size, whereas the number of repetitions and input and output channels is determined by the specific variant of the model. The segmentation model is just an extension of the detection model shown in Figure 2 that contains additional segmentation modules that are responsible for predicting segmentation masks.

### 2.6. Model Training

Before initiating the training, it is crucial to first complete tasks like preparing the imaging data sets using the GIMP program, implementing complex data augmentation and conducting image annotation. Having a well-prepared and annotated dataset is crucial to ensure the accuracy and effectiveness of the training process. The process of model training was a collaborative effort with computer engineers working closely alongside a radiologist and an oral and maxillofacial surgeon. Of all images, 60% were used for training, 20% for validation and 20% for testing. The model underwent fine-tuning with original image dimensions of 2776 × 1480, using stochastic gradient descent with a learning rate of 0.01. The training spanned across 100 epochs with a batch size of 4, which was the maximum possible size considering the experimental conditions. A grid search was conducted to find the best combination of augmentation methods and model variants. Experiments were conducted using a NVIDIA RTX A6000 (NVIDIA, Santa Clara, CA, USA) graphics card. The time required for one epoch was 40 s and inference after training was around 60 ms per sample.

### 2.7. Performance Evaluation Method

Performance of the model was evaluated using Intersection over Union (IoU), precision, recall, mAP@50 and mAP@50-95. They are defined by the following formulas:(1)$$precision=\frac{true\;positive}{true\;positive+false\;positive}$$
(2)$$recall=\frac{true\;positive}{true\;positive+false\;negative}$$
(3)$$mAP=\frac{1}{C}\sum_{n=1}^{C}AP$$

Precision, Equation (1), is the ratio of true positive predictions to the total number of positive predictions made by the model, while recall, Equation (2), is the ratio of true positive predictions to the total number of actual positive instances in the dataset. Average precision is defined as the area under the precision–recall curve, while mean average precision, Equation (3), is then calculated as a mean of average precisions across all classes. Whether an example is classified as true positive or not is defined by the parameter IoU. If the area of overlap between the real bounding box or a mask and the predicted one is higher than the threshold value determined by IoU, an example is classified as true positive. For recall and precision, IoU was set at 0.45, and for mAP@50 at 0.5. The mAP@50-95 value was calculated by averaging the AP values across different IoU thresholds (0.5 to 0.95 in increments of 0.05).

## 3. Results

The test set performance of the trained model is presented in Table 2 and Table 3. The results demonstrate a substantial enhancement across all evaluation metrics for both tasks due to the use of augmentation methods. The precision–recall (PR) curves for the detection and segmentation of lesions are shown in Figure 3.

In the detection task, the precision, recall, mAP@50 and mAP@50-95 scores without augmentation were recorded at 91.8%, 57.1%, 75.8% and 47.3%, respectively. However, by utilizing augmentation techniques, these figures experienced enhancement, with precision, recall, mAP@50 and mAP@50-95 reaching 95.2%, 94.4%, 97.5% and 68.7%, respectively. Similarly, in the segmentation task, the precision, recall, mAP@50 and mAP@50-95 values achieved without augmentation were 76%, 75.5%, 75.1% and 48.3%, respectively. However, the application of augmentation methods led to an improvement of these scores to 100%, 94.5%, 96.6% and 72.2%, respectively. Additionally, after finding the optimal hyperparameter configuration, we conducted another training session using five-fold cross-validation to mitigate the limitations associated with our relatively small dataset. The dataset was divided into five different train and validation sets, with each pair of sets used for model training. Results averaged across all five training iterations and are summarized in Table 4.

In the detection task, the precision, recall, mAP@50 and mAP@50-95 scores were recorded at 94%, 86.5%, 92.5% and 66.7% respectively. Similarly, in the segmentation task, the precision, recall, mAP@50 and mAP@50-95 values achieved were 88.3%, 87.4%, 94.15% and 66.4% respectively.

The trained model is capable of predicting bounding boxes, segmentation masks and probabilities, as shown in Figure 4 and Figure 5.

In addition to evaluating the model’s performance on cases with lesions, we also conducted tests on a separate set of 100 orthopantomograms devoid of lesions and without any other underlying medical issues. Remarkably, the model demonstrated impeccable performance on this set, yielding no false positives, affirming its robustness in accurately identifying the absence of anomalies in such cases.

## 4. Discussion

Deep learning models, particularly CNNs, have demonstrated remarkable performance in various radiological tasks, ranging from image segmentation and detection to disease classification [18,19]. These models excel in capturing complex patterns and variations that are often challenging for human observers [20]. The performance of deep learning models heavily relies on high-quality and well-labeled training data. Access to diverse and comprehensive datasets is crucial to ensure robust and generalizable models. Deep learning has emerged as a transformative force, revolutionizing various aspects of dental diagnosis, treatment planning, and patient care. Leveraging the capabilities of artificial intelligence and neural networks, deep learning technologies are reshaping the field by offering sophisticated solutions for improved accuracy and efficiency.

Lower jaw lesions comprise a broad spectrum of lesions, and they can arise from remnants of the odontogenic epithelium entrapped in bone or gingival tissue or develop from the epithelium of non-odontogenic origin [21]. Odontogenic lesions usually occur in relation to one tooth or a component of a tooth while non-odontogenic lesions usually have no relation with teeth or may involve a large part of the bone near two or more teeth [22].

Deep learning algorithms can identify subtle signs of dental diseases at an early stage, enabling timely intervention and preventing disease progression. This has the potential to significantly improve patient outcomes. However, there are only a few studies that have described using deep learning in the interpretation of panoramic radiographs. One study evaluated the contact between the lower third molar and inferior alveolar nerve while another evaluated the detection and diagnosis of dental caries [15,23]. Extra roots of mandibular first molars and development stage of third molars were also examined using deep learning on panoramic radiographs [10,24].

We developed a new model for the detection and segmentation of radiolucent lesions of the lower jaw using the latest YOLOv8. YOLO is an object detection algorithm that has gained popularity due to its ability to perform real-time object detection in images and videos with impressive accuracy. YOLO divides the input image into a certain number of equal parts and performs predictions for each part individually. If the center of the object is located within a certain part, that part is responsible for predicting that object. One of the novelties introduced by YOLO is non-maximum suppression (NMS), which is used to identify and remove redundant predictions [25].

In this study, the resolution of the input panoramic images was 2776 × 1480 pixels. To the best of our knowledge, our study has the largest resolution of panoramic radiographs to date, among published deep learning studies on the automatic detection of radiolucent lesions [10,11,12]. The developed model based on YOLOv8 showed a detection precision of 0.918 without augmentation and 0.952 with augmentation. The study by Kwon et al. reported 0.87 precision [12]. Another study reported the sensitivity (recall) for the detection of mandibular radiolucent lesions of 0.88 [10]. In the mentioned paper, the total number of panoramic radiographs was 1282 which was increased 12-fold using data augmentation techniques [12]. Despite the fact we used 200 panoramic radiographs containing 226 lesions, our model, developed by using newer, state-of-the-art YOLOv8, showed better detection precision than previously developed models. Another study employed a pre-trained neural network, DetectNet, for detecting mandibular lesions. Their original training dataset consisted of 200 samples, with an additional 60 samples in the test dataset. Using a pre-trained network, they achieved high recall for Stafne’s bone cavity, but only 0.79 for other radiolucent lesions of the mandible [26]. In contrast, in our study, despite having a relatively small dataset, our detection recall reached 0.94. Similar to us, Poedjiastoeti et al. adopted a transfer learning approach with a 16-layer CNN (VGG-16) to address the challenge of limited patient data. This led to a detection recall of 0.81 [18]. Furthermore, to improve our model’s generalization capabilities, we conducted five-fold cross-validation. In the detection task, the precision, recall, mAP@50 and mAP@50-95 values achieved were 0.94, 0.865, 0.925 and 0.667, respectively. As expected, there has been a slight drop in all metrics, which is attributed to the increased data diversity. Nonetheless, this has resulted in an improvement in the model’s overall generalization capabilities.

Additionally, in this study, we also performed complex lesion segmentation tasks which none of the previously published works have researched. Segmentation plays a central role in a broad range of applications across various fields, including medical image analysis. It is the process of dividing an image or data into distinct parts or regions for analysis. Segmentation is a crucial task in computer vision that enables machines to understand and interpret visual information with remarkable precision. There are various types of segmentation tasks, such as semantic segmentation, instance segmentation and panoptic segmentation. Semantic segmentation delineates each pixel in an image with a specific class label, allowing for a pixel-level understanding of the scene. Instance segmentation goes further by not only categorizing pixels into classes but also distinguishing between individual objects of the same class, assigning them unique identities, while panoptic segmentation unifies both semantic and instance segmentation [27].

Our model with augmentation techniques, in segmentation tasks, showed a precision of 1, recall of 0.945, mAP@50 of 0.966 and mAP@50-95 of 0.722. When applying five-fold cross-validation, the precision, recall, mAP@50 and mAP@50-95 values achieved were 88.3%, 87.4%, 94.15% and 66.4%, respectively.

Several limitations are inherent in this study. Deep learning, as employed here, requires a large amount of labeled training data. Our dataset comprises various types of lesions with radicular cysts being the predominant category. Despite our utilization of advanced data augmentation methods and the state -of-the-art YOLOv8 model, it is worth acknowledging that performance improvements could be achieved with a larger, more diverse dataset. Further research should also be focused on not only detecting but also classifying lesions using the YOLOv8 model.

Furthermore, it is important to note that our study was limited to radiolucent lesions of the lower jaw. Future research could aim to broaden the applicability of YOLOv8 to the upper jaw, thereby expanding its potential utility.

Lastly, this study is based on 2D panoramic radiographs, while there is a progression of using 3D cone beam computed tomography (CBCT) with the ability to produce volumetric images of the jaw bone at a reasonable cost and radiation dose. It is valuable to surgeons in pathology assessment because it provides horizontal, vertical, and axial views of structures. We fully recognize the potential benefits of extending our research to incorporate 3D models and CT scans. Further research should explore the application of our model in a 3D context.

Models developed through this approach possess the capability to significantly enhance diagnostic accuracy, streamline workflow and ultimately contribute to improved patient outcomes. By aiding radiologists in identifying subtle abnormalities and reducing interpretation time, deep learning models possess the potential to completely revolutionize the fields of radiology, dentistry and oral and maxillofacial surgery.

## 5. Conclusions

Our study confirmed that the model developed using state-of-the-art YOLOv8 is proficient in automatically detecting and segmenting radiolucent lesions within the mandible, exhibiting remarkable performance with high evaluation metrics. With its continual evolution and integration into various medical fields, the deep learning model holds the potential to revolutionize patient care, ushering in an era where precision medicine becomes the norm rather than the exception. However, its full potential can only be realized through collaboration among healthcare professionals, data scientists, and ethical considerations to ensure its responsible and beneficial implementation.

## Figures and Tables

**Figure 1 medicina-59-02138-f001:**
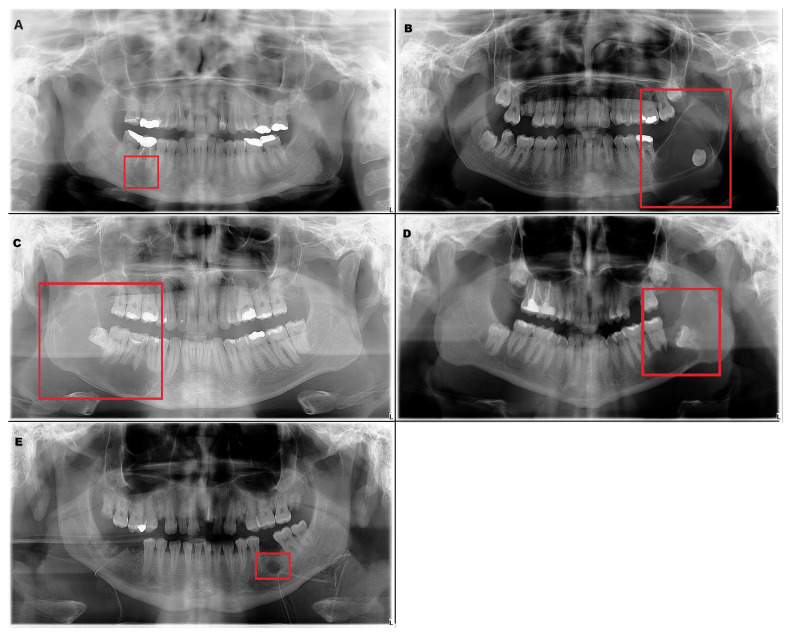
Examples of the included lesions. (**A**) Radicular cyst, (**B**) Ameloblastoma, (**C**) Odontogenic keratocyst (OKC), (**D**) Dentigerous cyst, (**E**) Residual cyst.

**Figure 2 medicina-59-02138-f002:**
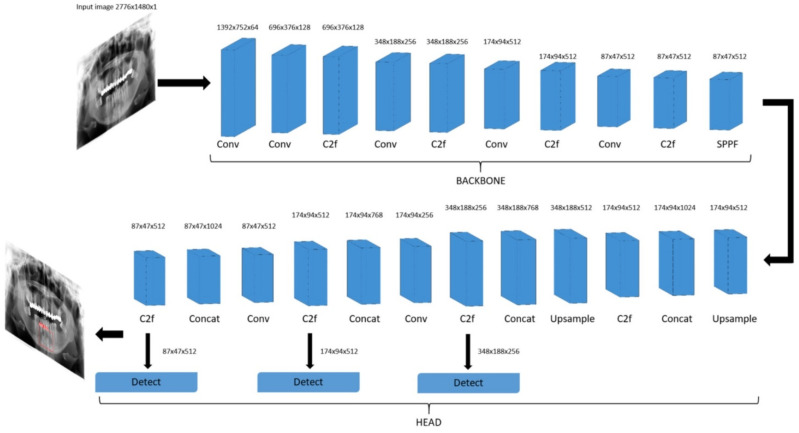
Schematic diagram of You Only Look Once (YOLOv8) detection model.

**Figure 3 medicina-59-02138-f003:**
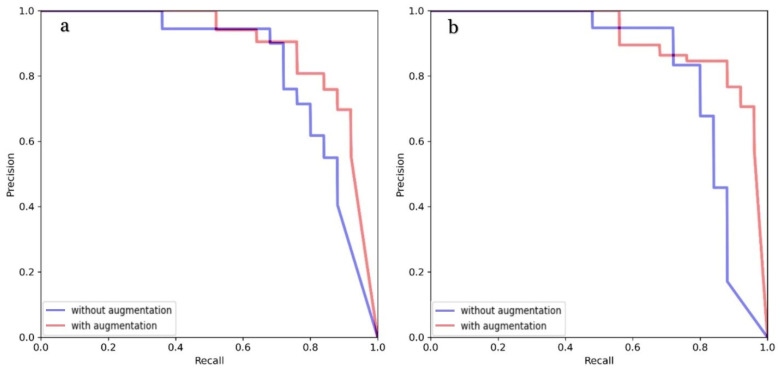
PR curves with and without augmentation for detection (**a**) and segmentation (**b**).

**Figure 4 medicina-59-02138-f004:**
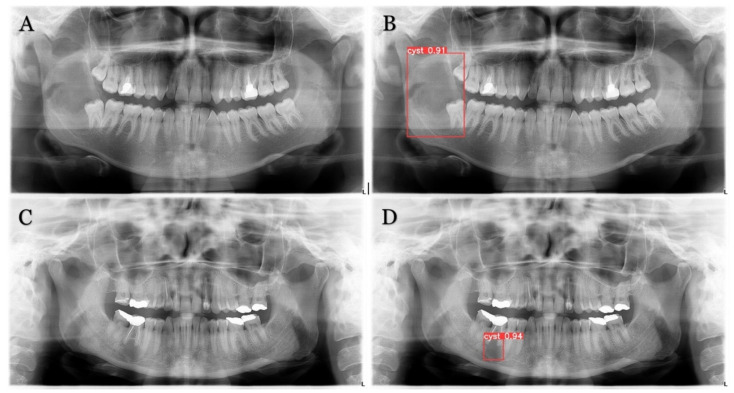
Model detection prediction of the radiolucent lesion in the lower jaw. (**A**), lesion in the ramus of the mandible (**B**), model successful detection of 91% (**C**), lesion in the corpus of the mandible (**D**), successful detection of 94%.

**Figure 5 medicina-59-02138-f005:**
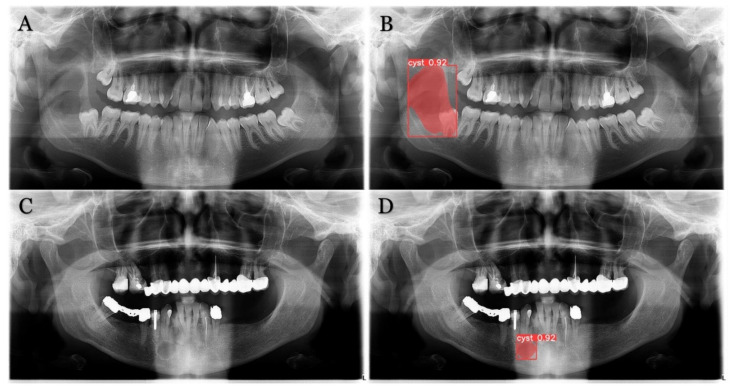
Model segmentation predictions of the lesions. Radiolucent mandibular lesions seen on panoramic radiographs shown on (**A**,**C**). Successful model segmentation with 92% shown on (**B**,**D**).

**Table 1 medicina-59-02138-t001:** Distribution of data.

Radiolucent Lesion	Number of Examples
Radicular cysts	138
Ameloblastomas	13
Odontogenic keratocysts	33
Dentigerous cysts	29
Residual cysts	13
**TOTAL**	**226**

**Table 2 medicina-59-02138-t002:** Performance of the model in the detection of radiolucent lesions; precision (positive predictive value), recall (sensitivity), mean average precision.

	Precision	Recall	mAP@50	mAP@50-95
without augmentation	0.918	0.571	0.758	0.473
with augmentation	0.952	0.944	0.975	0.687

**Table 3 medicina-59-02138-t003:** Performance of the model in the segmentation of radiolucent lesions; precision (positive predictive value), recall (sensitivity), mean average precision.

	Precision	Recall	mAP@50	mAP@50-95
without augmentation	0.76	0.755	0.735	0.483
with augmentation	1	0.945	0.966	0.722

**Table 4 medicina-59-02138-t004:** Performance of the model after using five-fold cross-validation.

	Precision	Recall	mAP@50	mAP@50-95
detection	0.94	0.865	0.925	0.667
segmentation	0.883	0.874	0.941	0.664

## Data Availability

The data presented in this study are available on request from the corresponding author. The data are not publicly available due to privacy restrictions.

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
