# Peer review of "Detection and Segmentation of Radiolucent Lesions in the Lower Jaw on Panoramic Radiographs Using Deep Neural Networks"

_medicina, 2023, doi:10.3390/medicina59122138_

Round 1

Reviewer 1 Report

Comments and Suggestions for Authors

Dear editor and author,

This well-prepared study aims to detect and segment radiolucent lesions in the lower jaw on panoramic radiographs using deep neural networks.  It presents a clear method and adequate data processing for a definite outcome. Moreover, it is noted that the manuscript still needs modifications. Here some significant flaws are as follows:

Comments 1:

How is the sample size calculation of this research?  Is the sample size enough to construct an AI-based model to verify the disease and region of the tumor?

Comments 2:

The number of the disease variety severely differ, as shown in Table 1, how can the author promise the accuracy of the model concerning the small sample size (13 ameloblastomata, residual cysts)?

Comments 3:

This study is based on 2D panoramic and the authors try to detect the region of the lesion, however, the 3D region of the tumor is often much more significant to the surgeon, how do the authors consider that question, do they consider applying 3D CT to this model? Do the model and result also can be applied in 3D models?

Comments on the Quality of English Language

Dear editor and author,

This well-prepared study aims to detect and segment radiolucent lesions in the lower jaw on panoramic radiographs using deep neural networks.  It presents a clear method and adequate data processing for a definite outcome. Moreover, it is noted that the manuscript still needs modifications. Here some significant flaws are as follows:

Comments 1:

How is the sample size calculation of this research?  Is the sample size enough to construct an AI-based model to verify the disease and region of the tumor?

Comments 2:

The number of the disease variety severely differ, as shown in Table 1, how can the author promise the accuracy of the model concerning the small sample size (13 ameloblastomata, residual cysts)?

Comments 3:

This study is based on 2D panoramic and the authors try to detect the region of the lesion, however, the 3D region of the tumor is often much more significant to the surgeon, how do the authors consider that question, do they consider applying 3D CT to this model? Do the model and result also can be applied in 3D models?

Author Response

Dear reviewer,

Thank you for your insightful comments and valuable feedback on our paper. We appreciate your engagement with our work and the opportunity to address your queries

  1. We acknowledge that our sample size is limited to 226 lesions, which was the maximum number we could collect within the scope of our study. Despite the relatively small sample size, we are encouraged by the robustness of our results. In particular, our AI-based model demonstrated consistent performance, achieving remarkable results on the test set (Precision of 95.2% with augmentation). Additionally, when applied to medical images without any discernible medical conditions, the model exhibited no false positives or other erroneous detections.
  2. Thank you for your for bringing attention to the variability in disease types, as highlighted in Table 1. In this particular research, our primary focus was on the general detection and segmentation of cysts in panoramic images, rather than emphasizing the specific diagnosis of cyst types. We acknowledge the variation in lesion types within our dataset, including instances where certain types have a smaller representation. That is why we also performed five fold cross validation which helped in reducing the variance and provided a more reliable estimate of the model's performance. Precision of  94% and recall of 86.5%  in detection task showed model’s overall generalization capabilities.
  3. Regarding the consideration of 3D regions in our study on detecting lesions in 2D panoramic images. We appreciate your  observation and welcome the opportunity to address this aspect of our research. We acknowledge the significance of 3D information, especially in the context of surgical planning where the  extent of tumors is crucial for surgeons. Regrettably, due to the constraints of our current study, we did not have access to 3D images, and our focus remained on the detection and segmentation of lesions in 2D panoramic images. We fully recognize the potential benefits of extending our research to incorporate 3D models and CT scans. To adapt our model based on YOLOv8  for 3D tasks, we need modifications and integrations with 3D reconstruction or depth estimation to accurately detect objects in a 3D space. Expanding our model to include 3D images is a logical progression for future investigations and  we plan to incorporate 3D imaging into our research.

Your diligence in reviewing our work is genuinely appreciated, and your constructive feedback will shape our future research

Reviewer 2 Report

Comments and Suggestions for Authors

Thank you very much for the article. 'Odontogenic lesions..'

in paragraph 3 of the introduction and 'Cysts...' sentences can be removed. Those who read the article are already familiar with the diagnoses.

In Figure 1, the lesions are in white boxes with thicker edges should be shown. This current figures are not clear.

Why image dimensions are selected as 2776 x 1480 and should be explained more clearly in the materials and method section.

In Table 2 does not need the horizontal line in the first column.

BW,

Author Response

Dear reviewer,

We wish to thank you for all the time and effort that you have put into reviewing this paper and we are highly grateful for your comments.

  • We will delete paragraph 3 in introduction about cysts and lesions 
  • We agree that Figure 1 is not very clear and will try to make the resolution much better
  •  Our focus has been on the detection and segmentation of lesions which is a complicated procedure that has not yet been discussed in previously published papers. That is why we needed highest possible resolution to attack problem of segmentation. Image dimensions are selected as 2776 x 1480 PPI because it was the highest resolution possible made by our Panoramic X-ray. As you suggested, we will describe it in the Materials and methods.
  • We will delete the horizontal line in the Table 2. Its not visible in pdf version, only in word and we missed it.

We would like to express our gratitude once again for all your comments.

Round 2

Reviewer 1 Report

Comments and Suggestions for Authors

Dear authors, 

Thank you for the modification. I still have some concerns about your answers.

Firstly, how can the generalisability of the model be ensured when masses may have different boundaries and densities due to pathology differences, which can lead to image consistency?

Also, how is the model compared with others' similar studies,  what are aspects of this study that are superior to other studies, as your recall rate is 86% less than 90%? How is the result compared with other studies, and how can you improve the precision of the model? 

Comments on the Quality of English Language

Dear authors, 

Thank you for the modification. I still have some concerns about your answers.

Firstly, how can the generalisability of the model be ensured when masses may have different boundaries and densities due to pathology differences, which can lead to image consistency?

Also, how is the model compared with others' similar studies,  what are aspects of this study that are superior to other studies, as your recall rate is 86% less than 90%? How is the result compared with other studies, and how can you improve the precision of the model? 

Author Response

Dear reviewer, 

  1. Thank you for raising an important concern regarding the generalizability of our model. It's crucial to clarify that our approach intentionally incorporates diverse training samples encompassing different boundaries and densities. The goal is not to enforce image consistency but rather to empower our model to recognize variations in pathology presentations, including differences in density and size. By exposing the model to a wide range of cases during training, we aim to enhance its adaptability to diverse scenarios. However, we acknowledge the ongoing nature of this research and are actively working on further refining our model's ability to generalize across varied pathology presentations. Additionally, we plan to expand our dataset to include a more comprehensive representation of different masses, ensuring the robustness of our model's generalizability. Your feedback is valuable in guiding our efforts towards achieving a more versatile and effective model.
  2. We appreciate your engagement and the opportunity to address your queries. Comparing our model with other similar studies can be challenging due to the inherent differences in datasets, models, and evaluation metrics employed across different research endeavors. Each study is conducted under unique circumstances that may affect outcomes. Nevertheless, to the best of our knowledge, none of the studies that are comparable to ours did a segmentation of lesions , neither  the five-fold cross validation that could show the real potential of the model.  They published research of lesions detection and showed  results with augmentation. Ariji Y, et al reported recall of 0.88, Kwon et al 0.87 and Poedijastoeti et al. 0.81, Recall metric, in our paper, in, detection mode using augmentation without five fold cross validation is 0.944. Moreover, we have incorporated the mean average precision as an additional metric in our evaluation. This metric is widely acknowledged as one of the best indicators of model performance in tasks involving detection and segmentation. Its inclusion provides a comprehensive view of our model's capabilities beyond a single metric. Lastly, we used a YOLOv8 which is the state of the art object detection.

      Future iterations will leverage larger datasets, incorporating the valuable insights gained from the experience of developing that was presented in this paper.